# The Gut Microbial Metabolite Pyrogallol Is a More Potent Inducer of Nrf2-Associated Gene Expression Than Its Parent Compound Green Tea (-)-Epigallocatechin Gallate

**DOI:** 10.3390/nu14163392

**Published:** 2022-08-18

**Authors:** Chen Liu, Sjef Boeren, Ignacio Miro Estruch, Ivonne M. C. M. Rietjens

**Affiliations:** 1Tea Refining and Innovation Key Laboratory of Sichuan Province, College of Horticulture, Sichuan Agricultural University, Chengdu 611130, China; 2Division of Toxicology, Wageningen University and Research, 6708 WE Wageningen, The Netherlands; 3Laboratory of Biochemistry, Wageningen University and Research, 6708 WE Wageningen, The Netherlands

**Keywords:** EGCG, Nrf2, proteomics, pyrogallol, t-BHQ

## Abstract

(-)-Epigallocatechin gallate (EGCG) has been associated with multiple beneficial effects. However, EGCG is known to be degraded by the gut microbiota. The present study investigated the hypothesis that microbial metabolism would create major catechol-moiety-containing microbial metabolites with different ability from EGCG to induce nuclear factor-erythroid 2-related factor 2 (Nrf2)-mediated gene expression. A reporter gene bioassay, label-free quantitative proteomics and reverse transcription quantitative real-time polymerase chain reaction (RT-qPCR) were combined to investigate the regulation of Nrf2-related gene expression after exposure of U2OS reporter gene or Hepa1c1c7 cells in vitro to EGCG or to its major microbial catechol-moiety-containing metabolites: (-)-epigallocatechin (EGC), gallic acid (GA) and pyrogallol (PG). Results show that PG was a more potent inducer of Nrf2-mediated gene expression than EGCG, with a 5% benchmark dose (BMD_5_) of 0.35 µM as compared to 2.45 µM for EGCG in the reporter gene assay. EGC and GA were unable to induce Nrf2-mediated gene expression up to the highest concentration tested (75 µM). Bioinformatical analysis of the proteomics data indicated that Nrf2 induction by PG relates to glutathione metabolism, drug and/or xenobiotics metabolism and the pentose phosphate pathway. Taken together, our findings demonstrate that the microbial metabolite PG is a more potent inducer of Nrf2-associated gene expression than its parent compound EGCG.

## 1. Introduction

Green tea is one of the most popular beverages enjoyed both by Asian and Western cultures. Habitual consumption of green tea has been reported to have a variety of health benefits [1,2]. These health-promoting properties are believed to be associated with the most abundant and bioactive green tea catechin, (-)-epigallocatechin gallate (EGCG).

Upon ingestion of green tea or EGCG, only a small portion (less than 1%) [3,4] of EGCG appears in the systemic circulation, while the majority reaches the large intestine, where it is degraded by intestinal microbiota (Figure 1) [5]. The microbial conversion starts with the rapid degalloylation of the D ring by microbial esterases, which gives rise to gallic acid (GA) and (-)-epigallocatechin (EGC). Subsequently, these two metabolites are subject to further degradations. GA produces pyrogallol (PG) by decarboxylation, and PG can be further degraded into catechol, butyric acid and acetic acid [6,7]. Previous anaerobic fecal incubations of EGCG revealed EGC to be formed at a maximum level of 16.4% of the original EGCG [6]. Subsequent reductive cleavage in the C ring of EGC gives rise to the formation of 1-(3′,4′,5′-trihydroxyphenyl)-3-(2″,4″,6″-trihydroxyphenyl)-2-propanol, which can be degraded to phenyl-γ-valerolactones [6,8,9,10]. The A-ring fission, lactonization, dehydroxylation, decarboxylation and beta-oxidation catalyzed by the gut microbiome results in the formation of various smaller phenolic acids, which are more readily absorbed [5,11]. It may be hypothesized that the above metabolic pathways not only improve the overall bioavailability of EGCG—or rather its metabolites—but may also contribute to the physiological bioactivities attributed to the parent compound. Differences in the chemical structure, kinetics and or cellular redox behavior may affect the Nrf2 induction potential of the catechols [12,13,14]. Therefore, it is of interest to characterize the bioactivities of EGCG and its microbial metabolites when investigating the modes of action behind the observed physiological effects induced by green tea or EGCG.

Among the cellular signaling pathways, the Kelch-like ECH associating protein 1/nuclear factor erythroid 2-related factor 2 (Keap1/Nrf2) system is often referred to as a pivotal pathway with vital regulatory roles in many physiological processes [15]. Studies have confirmed that several green tea catechins are able to facilitate the nucleus translocation of Nrf2 [16,17]. As a result, this transcription factor, Nrf2, subsequently binds to the electrophile responsive element (EpRE) in the promotor region of a wide range of Nrf2 responsive genes, initiating the cellular defense system by transcription of a wide array of cytoprotective genes, including glutathione S-transferases (GSTs), glutathione reductases (GRs) and UDP-glucuronosyltransferases (UGTs), among many others [15,18].

The activation of the Nrf2 pathway is believed to be able to protect cells from both oxidative and electrophilic stresses that originate from either exogenously or endogenously formed reactive oxygen species and/or electrophiles [19,20]. The aim of the present study was to characterize the bioactivity of the microbial metabolites formed and to compare these activities to those of the parent catechin taking EGCG as the model compound. The microbial EGCG metabolites to be tested in the present study were chosen based on: (i) abundance in their formation during the first 6 h anaerobic fecal incubation of EGCG [6] and (ii) the presence of a catechol moiety—the latter because such a structural element is known to be essential for efficient Nrf2 activation potency [16,21,22]. This second criterion in particular was not fulfilled for 5-(3′,5′-dihydroxyphenyl)-γ-valerolactone, the dominant EGCG metabolites in anaerobic fecal incubations with EGCG [6]. GA—which does contain a catechol moiety—was abundantly quantified in the earlier hours of in vitro anaerobic fecal incubations of EGCG, with a peak occurrence amounting to 33.9% (mol/mol) of the initial amount of EGCG [6]. GA was further decarboxylated to PG [5,23], which was quantified in a substantial amount from 3 to 6 h in in vitro anaerobic fecal incubations of EGCG, reaching a plateau at 31.7% of the EGCG added to the incubation [6]. PG was also reported to be one of the main metabolites in urinary excretion after green tea consumption by human volunteers [24].

The activity of EGCG and several of its selected catechol-moiety-containing microbial metabolites, including EGC, GA and PG, were quantified in an Nrf2 reporter gene assay by reverse transcription quantitative real-time polymerase chain reaction (RT-qPCR) and a label-free quantitative proteome analysis of exposed U2OS cells or Hepa1c1c7 cells. The results obtained provided valuable leads on the importance of intestinal microbial metabolites of EGCG regarding the activation of the Nrf2-mediated cellular defense system.

## 2. Materials and Methods

### 2.1. Chemicals and Reagents

EGCG, EGC, GA, PG, tert-butylhydroquinone (t-BHQ), l-ascorbic acid and curcumin were ordered from Sigma-Aldrich (Zwijndrecht, The Netherlands). Trypsin, penicillin/streptomycin (P/S), geneticin (G148) and non-essential amino acids (NEAA) were obtained from the Invitrogen Corporation (Breda, The Netherlands). Dimethyl sulfoxide (DMSO) was purchased from Acros Organic (Fair Lawn, NJ, USA). Fetal bovine serum (FBS) was bought from Capricorn Scientific (Ebsdorfergrund, Germany). Minimum Essential Medium (MEM) Alpha, Dulbecco’s Modified Eagle Medium with 1:1 (*v/v*) F-12 Nutrient Mixture (DMEM/F-12) and phosphate-buffered saline (PBS) were supplied by Gibco (Paisley, UK).

### 2.2. Cell Lines

The U2OS-Nrf2 CALUX cells (Biodetection Systems, Amsterdam, The Netherlands) were derived from human osteoblastic cells, transfected with a reporter construct carrying a luciferase reporter gene under transcriptional control of four different EpRE sequences [25]. This osteoblast cell line is considered to be an “empty” cell line in terms of endogenous receptors and is therefore generally and often used as the preferred cell line to create reporter gene assays, forming the basis for a whole range of reporter cell lines (https://biodetectionsystems.com (accessed on 1 June 2021)). The cells were cultured with DMEM/F-12 culture medium, supplemented with 1% (*v*/*v*) P/S, 1% (*v*/*v*) NEAA and 10% (*v*/*v*) FBS. The U2OS-Cytotox CALUX cells (Biodetection Systems, Amsterdam, The Netherlands) carry a luciferase reporter gene under transcriptional control of a constitutive promoter. As a result, the cells are able to produce an invariant luciferase expression. The Hepa1c1c7 cell line is a murine hepatoma cell line and was cultured in MEM-Alpha supplemented with 1% (*v*/*v*) P/S and 10% (*v*/*v*) FBS. This cell line was selected as a primary systemic target for exposure to EGCG and its intestinal metabolites and because it has been reported in several animal studies that green tea extract (GTE) or green tea polyphenols could exert hepatoprotective effect by inducing Nrf2-dependent genes expression (e.g., glutamate-cysteine ligase (Gcl), Heme oxygenase-1 (Hmox-1)) [8,19]. Controlled culturing conditions were provided, as previously established [6]. An amount of 200 µg/mL G418 was added to the culture medium once a week in order to maintain the selection pressure for the aforementioned cells lines.

### 2.3. Reporter Gene Assay

The bioactivity of EGCG, EGC, GA, PG and t-BHQ in inducing Nrf2 gene transcription was checked by measuring the luciferase activity in the U2OS-Nrf2 CALUX cells [25]. All assay medium contained freshly made 0.5 mM l-ascorbic acid to inhibit the auto-oxidation of the test compounds [26,27,28]. A total of five exposure concentrations were applied for all compounds, ranging from 5 µM to 75 µM (added from 200-times-concentrated stock solutions in DMSO). An amount of 20 µM curcumin served as a positive control.

To investigate the cytotoxicity, which could cause false negative results, and whether luciferase stabilization (false positive results) occurred during the exposure to the test compounds, a parallel U2OS-Cytotox CALUX assay was performed. This assay was performed using the same conditions as described above for the U2OS-Nrf2 CALUX assay. The results of both assays are presented as the induction factor (IF) compared to the solvent control (0.5% DMSO in exposure medium). Results from at least three independent biological replicates were obtained for all compounds.

### 2.4. Cell Viability Assay

The WST-1 assay was applied to assess the viability of Hepa1c1c7 cells after exposure to the compounds tested. Exposure concentrations leading to viabilities higher than 80% are considered non-cytotoxic. In brief, the cells were seeded in the inner 60 wells of a transparent 96-well plate with a density of 2 × 10^4^ cells per well in 100 µL culture medium. The subsequent exposure procedure was the same as described above for the CALUX assay. For the WST-1 assay, 6 µL WST-1 solution was transferred into each exposure well followed by 60 min incubation at 37 °C. The amount of formazan formed from WST-1 directly correlated to the number of viable cells and was quantified by measuring the absorbance at 440 and 620 nm with a microplate reader. The result was calculated by using the values at 440 nm to subtract their corresponding values at 620 nm. The viability (%) of the solvent control was set as 100%.

### 2.5. Sample Preparation and Liquid Chromatography–Mass Spectrometry (LC-MS) Analysis for Label-Free Quantitative Proteomics

To prepare cell samples, Hepa1c1c7 cells were seeded at a density of 6 × 10^5^ cells/mL in T25 flasks one day before exposure. Subsequently, cells were exposed to an assay medium containing 30 µM EGCG, EGC, GA, PG or t-BHQ for 24 h in the presence of 0.5 mM l-ascorbic acid. The concentration of the test compounds used was shown not to cause a significant decrease in cell viability (Appendix A). Afterward, cells were washed twice with PBS, and 1.5 mL 100 mM Tris-hydrochloride (Tris-HCl), pH 8, was added into the flask. The cells were then scraped and homogenized, then centrifuged for three minutes at 9391× *g*. The supernatants were discarded, and cell pellets were washed twice with 1 mL 100 mM Tris-HCl, pH 8. In the end, the cell pellets were sonicated and dissolved in 100 µL 100 mM Tris-HCl, pH 8. The concentration of protein in the samples was determined by the Bicinchoninic Acid (BCA) method [29]. To obtain the peptide samples, a total of 100 µg protein from each sample was subjected to sample preparation via the protein aggregation capture (PAC) method [30]. A final concentration of 0.1 µg/µL peptides was prepared for all samples, and these samples were stored immediately at −20 °C until further analysis. Four independent biological replicates were collected for all treatments.

An amount of 3 µL of peptide samples was loaded directly onto a 0.10 mm × 250 mm ReproSil-Pur 120 C18-AQ (Dr. Maisch, Ammerbuch, Germany) 1.9 µm beads analytical column (prepared in-house) at a constant pressure of 825 bar (flow rate of circa 600 nL/min) with buffer 1 mL/L formic acid in water and eluted at a flow of 0.5 µL/min with a 50 min linear gradient from 9% to 34% acetonitrile in water with 1 mL/L formic acid with a Thermo EASY nanoLC1000 (Thermo Fisher Scientific, Waltham, MA, USA). An electrospray potential of 3.5 kV was applied directly to the eluent using a stainless-steel needle fitted into the waste line of a micro cross that was connected between the nanoLC1000 and the analytical column. On the connected Orbitrap Exploris 480 (Thermo electron, San Jose, CA, USA), mass spectrometry (MS) and MSMS automatic gain control (AGC) targets were set to 300%, 100%, respectively, or maximum ion injection times of 50 ms (MS) and 30 ms (MSMS) were used. Higher energy collisional dissociation (HCD) fragmented (isolation width 1.2 m/z, 28% normalized collision energy) the MSMS scans in a cycle time of 1.1, recording the most abundant 2–5+ charged peaks in the MS scan in a data-dependent mode (Resolution 15,000, threshold 2e4, 15 s exclusion duration for the selected m/z +/− 10 ppm).

For identification and relative quantification of the peptides, the MaxQuant software package (1.6.3.4) (Max Planck Institute of Biochemistry, Planegg, Germany) was used. Data were filtered to show only the proteins that were reliably identified by at least two peptides, at least one of which was unique, and one was unmodified and had a false discovery rate (FDR) of less than 1% on both the protein and peptide level. Reversed hits were deleted from the MaxQuant result table. Zero label-free quantitation (LFQ) intensity values were replaced by a value of 10^6.5^ (a value slightly lower than the lowest measured value) to make sensible ratio calculations possible.

### 2.6. Data Analysis and Bioinformatics

After obtaining the whole protein dataset of all groups, a manual search for Nrf2-related proteins was conducted, based on recent publications [15,18,31,32,33,34]. The protein fold change (FC) was calculated as the ratio of the average LFQ intensities of a specific treatment and the control. The Nrf2-related proteins thus identified were derived and, together with their respective logarithmized LFQ intensities or FCs for the different treatments, were further subjected to the principal component analysis (PCA) with the help of R package ropls (R Core Team, Vienna, Austria). The hierarchic clustering of the treatments was visualized in a heatmap by ClustVis (https://biit.cs.ut.ee/clustvis/ (accessed on 1 April 2021)). Furthermore, R 3.6.0 (R Core Team 2019, Vienna, Austria) was used to perform a Student’s *t*-test for analyzing the statistical significance of differences between the control and different treatments. The differentially expressed proteins (DEPs) in different treatments were selected based on the FCs (≥1.2 or ≤0.8) with a *p*-value of <0.05. In the next step, the DEPs in each treatment were used as inputs to perform the gene ontology (GO) enrichment analysis (biological process (BP), cellular component, (CC) and molecular function (MF)) and Kyoto Encyclopedia of Genes and Genomes (KEGG) pathway enrichment via the DAVID functional annotation tool (https://david.ncifcrf.gov/tools.jsp (accessed on 1 April 2021)). The derived results were subjected to Bioinformatics (http://www.bioinformatics.com.cn (accessed on 1 April 2021)) for visualization. Additionally, DEPs were also used for protein–protein interaction (PPI) network analysis, which was performed via the STRING database (https://string-db.org/ (accessed on 1 April 2021)), and the results thus obtained were visualized using Cytoscape 3.8.2 (Institute for System Biology, Seattle, WA, USA). Hub protein networks were further produced by the plug-in software MCODE (Samuel Lunenfeld Research Institute, Toronto, ON, Canada) in Cytoscape 3.8.2.

### 2.7. RT-qPCR

To study the Nrf2 activation at the RNA level, RT-qPCR was carried out. To do so, Hepa1c1c7 cells were seeded in a 6-well plate at a density of 2.4 × 10^5^ cells/mL in 2.5 mL culture medium per well and left to attach for 24 h followed by another 24 h of exposure to 30 µM EGCG, EGC, GA, PG or t-BHQ in the presence of 0.5 mM l-ascorbic acid. After exposure, cells were washed twice with ice-cold PBS and lysed with RLT lysis buffer (Qiagen, Venlo, The Netherlands). QIAshredder and RNeasy kits (Qiagen, Venlo, The Netherlands) were used to extract high-quality RNA from cell samples. Subsequently, the RNA concentrations, A260/A280 and A260/230, were quantified using Nanodrop One (Thermo Fisher Scientific, Waltham, MA, USA). Afterward, RNA was reverse transcribed into cDNA using the QuantiTect Reverse Transcription Kit (Qiagen, Venlo, The Netherlands). Gapdh was chosen as the reference gene. The relative transcription level of NAD(P)H dehydrogenase (quinone) 1 (Nqo1), Glutathione s-transferase A3 (Gsta3), Glutamate-cysteine ligase catalytic subunit (Gclc) and UDP glucuronosyltransferase family 1 member a6 (Ugt1a6) were checked by RT-qPCR using SYBR Green-based method (Qiagen, Venlo, The Netherlands). The Rotor-Gene 6000 cycler (Qiagen, Venlo, The Netherlands) recorded the cycle threshold (Ct) values of all samples, and these data were used to calculate the relative RNA level by the 2^−ΔΔCt^ method. The primers were commercial QuantiTect Primer Assays ordered from Qiagen (Venlo, The Netherlands), namely Mm_Gapdh_3_SG, Mm_Nqo1_1_SG, Mm_Gsta3_va.1_SG, Mm_Gclc_1_SG and Mm_Ugt1a6_1_SG.

### 2.8. Benchmark Dose Analysis

In order to obtain the in vitro concentration–response curves of EGCG and its microbial metabolites in inducing Nrf2 signaling, the concentrations of the model compounds used in the U2OS-Nrf2 CALUX assay ranged between 5 and 75 µM. The benchmark dose (BMD) analysis on the in vitro concentration–response curves of EGCG and PG obtained in the U2OS-Nrf2 CALUX reporter gene assay was performed using US EPA Benchmark Dose Software (BMDS3.2) (US EPA, Washington, DC, USA). The models were selected based on the *p* values and Akaike information criterion (AIC) values.

### 2.9. Statistical Analysis

Data are presented as mean ± standard error of the mean (SEM). All experiments mentioned in the present study were conducted in at least three independent biological replications. GraphPad Prism 5.0 (GraphPad Software, San Diego, CA, USA) was used to plot the line charts and histograms. One-way ANOVA was performed on the data of relative RNA levels in Hepa1c1c7 cells after exposure to the selected model compounds in RT-qPCR experiments. For the U2OS-Nrf2 CALUX reporter gene assay, two-way ANOVA was performed to evaluate the induction factors by the compound (EGCG or PG) and the exposure concentrations and their interactions. The data of both the U2OS-Nrf2 CALUX reporter gene assay and RT-qPCR analysis were evaluated for equal variances and homogeneity of variance using the Shapiro–Wilk and Levene tests, respectively. Logarithmic transformation or taking the reciprocal were performed for variables with unequal variance. Statistical significance was determined at *p*-value < 0.05.

## 3. Results

### 3.1. Activation of Nrf2-Mediated Luciferase Expression by EGCG and Its Metabolites

The concentration response results of luciferase activity induced by EGCG, EGC, GA, PG and t-BHQ are presented in Figure 2A. At 0–30 µM concentrations, PG and t-BHQ induced significant increases in luciferase expression. For instance, at 15 µM PG and t-BHQ, the induction factor (IF) values amounted to 2.1 and 1.9, respectively. At 30 µM PG and t-BHQ, the induction was 3.3- and 2.6-fold. On the other hand, EGCG was less active, with an IF of 1.0 and 1.6 at 15 and 30 µM, respectively. The two-way ANOVA of IFs induced by EGCG and PG further indicated both main effects of the compound and concentration significantly affected the IF values (*p* < 0.0001), while their interactive effect was not significant (*p* = 0.3531). Therefore, the data were corrected for Bonferroni multiple comparisons of the IF values between EGCG and PG treatments at the same exposure concentrations (Figure 2B). The metabolite PG was more capable of inducing Nrf2 activation than EGCG, especially at concentrations ranging between 15 and 50 µM (*p* < 0.05) (Figure 2B). GA and EGC were inactive at all the testing concentrations. The average IF of the positive control (20 µM curcumin) was 42.3, indicating the assay worked well. Benchmark dose modeling of the derived concentration–response data resulted in 5% benchmark dose (BMD_5_) values for EGCG and PG mediating the induction of Nrf2-mediated gene expression of 2.45 and 0.35 µM, respectively (Appendix A). From these concentrations onwards, the induction of Nrf2 can be expected. The results of the U2OS-Cytotox CALUX assays showed no luciferase stabilizations for any of the five compounds at any of the tested concentrations, except for EGCG, which showed a slight increase (1.7-fold) in luciferase activity at 50 and 75 µM (Figure 2C). The IFs for the Nrf2 CALUX assay at these two concentrations for EGCG were corrected for this response in the Cytotox CALUX assay. No cytotoxicity was found for EGCG, EGC, GA and PG up to the maximum concentration tested. However, the U2OS cells seemed to be much more susceptible to the cytotoxicity of t-BHQ, with IF dropping from 15 µM (0.87) and decreasing to 0.28 at 75 µM (Figure 2C).

### 3.2. Nrf2-Related Protein Expression in Hepa1c1c7 Cells

Based on the results shown in Figure 2A,B, at 30 µM, both PG and t-BHQ were able to induce an induction that was substantially above the 2-fold threshold, which was considered as a positive response in the U2OS-Nrf2 CALUX reporter gene assay. Therefore, to further study the changes of Nrf2-associated gene expression, 30 µM was selected as the concentration tested in the proteomics study. In total, 3973 proteins were detected in all samples, and their fold changes (FCs) compared to solvent control were calculated (Appendix A). Furthermore, a manual selection of the literature-reported Nrf2-related proteins provided a list of 97 proteins (Appendix A). The LFQ intensity values for these 97 proteins (in four biological replicates) were used as the input for PCA. The PCA explained 39% of the variation in the protein profiles with PC1 and PC2 (Figure 3A). PC1 accounted for a major part of the variation (23%) in the dataset, whereas PC2 captured 16% of the overall variation. On the PCA plot, EGCG, EGC, GA and the control groups located close to one another, indicating the inactivity of EGCG, EGC and GA in activating the Nrf2-mediated protein expression at 30 µM. In contrast, PG and t-BHQ treatments separated from the controls and the other treatments in the PCA graph, suggesting that these two treatments caused differential changes in the expression levels of Nrf2-related proteins. A PCA scatter plot using the mean values of FCs from four biological replicates can be found in Appendix A. The hierarchic clustering of the treatments based on the mean values of FCs in four replicates was visualized using a heatmap (Figure 3B). In agreement with the results of the PCA analysis, the PG and t-BHQ treatments separated from the EGCG, EGC, GA and the control groups. This was mainly due to the upregulation of proteins in both the PG and t-BHQ treatments in contrast to the non-activation in the other treatments. These upregulated proteins are presented in red color in the heatmap in Figure 3B. For instance, the FCs of Aldehyde dehydrogenase family 3, subfamily A1 (Aldh3a1, protein ID: P47739) in PG and t-BHQ treatments were 1.9 (*p* < 0.001) and 2.5 (*p* < 0.001), respectively, while they were 1.0 (all non-significant) in the other groups. The FCs of NAD(P)H dehydrogenase (quinone) 1 (Nqo1, protein ID: Q64669) in PG and t-BHQ treatments were 3.1 (*p* < 0.05) and 4.3 (*p* < 0.05), respectively, while they ranged from 0.9 to 1.1 (all non-significant) in EGCG, EGC, GA treated and the control groups. A heatmap plotted with logarithmized protein intensities of individual four replicates in each treatment can be found in Appendix A.

### 3.3. Differentially Expressed Proteins (DEPs) and Functional Enrichment Analysis

By comparing the treated groups with the solvent control group, a total of number of 213, 151, 180 and 227 DEPs were identified for EGCG, EGC, GA and PG treatments (Figure 4A). As the Nrf2 activator, the t-BHQ treatment had the highest number of DEPs: 419, including 161 upregulated and 258 downregulated DEPs. EGCG treatment resulted in 93 upregulated and 120 downregulated DEPs; EGC treatment resulted in 72 upregulated and 79 downregulated DEPs; GA treatment resulted in 93 upregulated and 87 downregulated DEPs; and PG treatment had 103 upregulated and 124 downregulated DEPs. Subsequently, the GO enrichment and KEGG pathway enrichment analyses were performed for DEPs detected in different treatments. The GO enrichment analysis consisted of three parts: BP, MF and CC. The enrichment results of the three parts were plotted as bubble charts, where the *x*-axis presents the fold enrichment and the *y*-axis the −log_10_ (*p*-value) of different annotations. The size of the circles positively correlated with the number of DEPs that fell into the term (Appendix A). In line with the results presented in Figure 3, the Nrf2-associated terms were only enriched in PG and t-BHQ treated cells. For instance, the glutathione metabolic process and the oxidation-reduction process were both observed upon PG and t-BHQ treatments in terms of BP. Oxidoreductase activity and glutathione binding were both obtained in PG and t-BHQ treatments for MF (Appendix A). Furthermore, the KEGG pathway analysis enabled a better understanding of the relationship among the different DEPs (Figure 4B,C and Appendix A). Figure 4B,C present the top ten most significantly enriched KEGG terms in PG and t-BHQ treated cells, respectively. Glutathione metabolism was the term that was enriched most significantly in both treatments. Moreover, the metabolism of xenobiotics by cytochrome P450 and drug metabolism-cytochrome P450, which are the second and third most significant pathways enriched in PG treatment, were also found in t-BHQ treated cells. These findings suggest that PG and t-BHQ shared some common characteristics regarding the glutathione synthesis or metabolism, which are the vital consequences of Nrf2 activation. The other compounds tested at 30 µM were not able to trigger the biological processes and pathways critical to Nrf2 signaling.

### 3.4. Protein–Protein Interaction (PPI) Network Analysis

Figure 5 shows the results of the PPI network analysis for the PG and t-BHQ treatments. The PPI results of the other treatments can be found in Appendix A. The PPI interaction network of the PG treatment comprised 175 nodes and 443 edges (Figure 5A). Meanwhile, as more DEPs were enriched after t-BHQ treatment, the PPI network of t-BHQ treatment resulted in 375 nodes and 1787 edges (Figure 5C). To further visualize the essential proteins among the complex PPI networks, the hub protein networks were produced via the MCODE plugin in the Cytoscape software. Several hub proteins were detected in both PG and t-BHQ PPI networks. These hub protein networks were ranked according to their network scores, which were calculated using the MCODE algorithms. The analysis revealed that nine out of ten nodes from the top one hub protein network of the PG treatment (Figure 5B) were also found in the second high-scored hub protein network of the t-BHQ treatment (Figure 5D), and all of these hub proteins were Nrf2-asscociated proteins. Specifically, ten nodes (Gclc, Gstm1, Gpx8, Gsr, Gsta3, Gsta4, Gstp1, Cryz, Gsto1 and Nqo1) connected by 42 edges are depicted in the PG hub protein network (Figure 5B). Fifteen nodes (Gclc, Gstm1, Gsr, Gsta3, Gsta4, Gstp1, Cryz, Gsto1, Nqo1, Gss, Txnrd1, Aldh3a, Mgst1, Cyp1a1 and Cat) connected by 85 edges are presented in the t-BHQ hub protein network depicted in Figure 5D. These data suggest that the Nrf2 signaling activation is an important response following exposure to PG or t-BHQ.

### 3.5. A Further Analysis of “Target” Proteins from PG and t-BHQ Treatments

Since the same KEGG terms (glutathione metabolism, metabolism of xenobiotics by cytochrome P450 and drug metabolism-cytochrome P450) were enriched both in PG and t-BHQ treatments, and given that the hub proteins were all Nrf2-related proteins, these proteins, together with their gene names and FCs, are summarized in Appendix A. In total, 22 proteins were obtained. In the EGCG, EGC and GA treatments, all these proteins remained unchanged, with protein intensity FCs ranging from 0.8 to 1.2. However, their FCs ranged from 0.8 to 3.1 in the PG group and from 0.8 to 4.3 in the t-BHQ group. PCA and hierarchic clustering analysis were processed based on the FCs of the 22 proteins in different treatments of Hepa1c1c7 cells. The PCA explained a total of 94.0% of the variation in the protein dataset with two PCs (PC1 and PC2) (Appendix A). PC1 accounted for the majority of the total variation (85.2%) in the dataset, whereas PC2 captured only 8.8% of the overall variation. From the PCA plot, it follows that the EGCG, EGC, GA and control groups were close to each other, while the PG treatments were separated far from them, and the t-BHQ group located even farther away. Meanwhile, on the heatmap (Appendix A), almost all proteins were upregulated in the PG and t-BHQ groups (except quinone oxidoreductase and probable glutathione peroxidase 8 in both groups and cytochrome P450 1A1 in the PG group) in contrast to the responses observed for all the proteins in the other exposure groups. This difference indicates that the PG and t-BHQ groups cluster together, while all the other groups are in another cluster. The findings here are in line with the analysis for the 97 Nrf2-related proteins in Section 3.2 but also show a more vivid contrast between the PG, t-BHQ treatments and the other groups. Taken all together, the results reveal that, at moderate concentrations, EGCG is not potent enough to trigger the Nrf2 signaling, while its microbial metabolite, PG, seems able to activate the cascade in Hepa1c1c7 cells.

### 3.6. RT-qPCR Analysis

To further corroborate the effects of PG, the Nrf2-mediated gene expression was investigated by RT-qPCR. Four well-known Nrf2-regulated genes were selected [15,34]. Their corresponding protein names, protein IDs and protein intensity FCs can be found in Appendix A. The RT-qPCR results are presented in Figure 6 and are in line with the proteomics data. Only PG- and t-BHQ-exposed Hepa1c1c7 cells showed upregulated (1.5–10.2-fold) gene expressions for all four genes, while gene expressions in the EGCG-, EGC- or GA-exposed cells were not affected (<1.5-fold) compared to the solvent control (Figure 6). Although the proteomics results and RT-qPCR results share increased gene and protein expression characteristics, it is worth mentioning that relative RNA levels do not always align exactly with the respective protein levels from the proteomics results in terms of the fold changes observed. For example, the relative RNA level of Gsta3 was 10.2 (Figure 6), while the FC of glutathione S-transferase A3 (the corresponding protein) amounted to 2.7. This can be due to, for example, the saturation in the mRNA translation efficiency and/or post-translational modifications via miRNAs, etc. Nevertheless, both our proteomics and RT-qPCR results corroborated the hypothesis that the microbial metabolite, PG, was more potent in inducing Nrf2 signaling compared to the parent compound EGCG.

## 4. Discussion

Employing an Nrf2 reporter gene assay, RT-qPCR and label-free quantitative proteomics, our current study characterized the differences between EGCG and its microbial metabolites, EGC, GA and PG, in inducing Nrf2-mediated gene expression both at the transcriptional and translational levels. The results indicate that both the parent compound EGCG and especially its catechol-moiety-containing metabolite PG are able to induce Nrf2-mediated gene expression. However, at the micromolar exposure concentration (e.g., 15 to 50 µM), especially PG, rather than EGCG, appeared able to alter Nrf2-related gene expression.

EGCG is one of the most popular polyphenols that is intensively investigated by numerous researchers due to its versatile pharmacological benefits. It accounts for more than 50% of the total catechin content in green tea [35]. As other flavonoids, EGCG has a molecular structure comprising two phenyl rings (A ring and B ring), which are connected by a heterocyclic ring (C ring). In addition, a galloyl moiety connects to the C ring in the 3-position, which is known as the D ring of EGCG. Therefore, EGCG has the highest number of (eight) free hydroxy groups compared to other catechins, which is often considered as the reason for its potent bioactivities [3,36]. For example, Dey and co-workers found EGCG being superior to (+)-catechin in alleviating high-fat-diet-induced non-alcoholic steatohepatitis in C57BL/6J mice at the same dose level [37]. Interestingly, in the present study, we revealed that PG, which is one of the major microbial metabolites formed from EGCG, both in in vitro and in vivo studies [6,24], exhibits a more potent ability in inducing the Nrf2 signaling pathway. It has been recognized that the parent compound EGCG possesses very limited transepithelial absorption and transport, and an obvious efflux is rather mediated by multidrug resistance proteins [5,38,39]. Additionally, Grzesik and colleagues analyzed the auto-oxidation of 54 dietary antioxidants. They found that both the EGCG and PG could generate hydrogen peroxide in DMEM medium, with PG being more potent than EGCG [28]. This might provide another explanation for the difference in the potency for Nrf2-mediated induction, since it suggests that the pro-oxidant activity, accompanied by the formation of reactive oxygen species (ROS) and reactive quinone metabolites, is more efficient with PG than EGCG. These products of pro-oxidant chemistry of the polyphenols, especially the ones with a catechol-type moiety [16,21,22,36], could either directly react with cysteine residues of Keap1, causing the dissociation of Nrf2, or conjugate with reduced glutathione (GSH), thereby leading to cellular oxidative stress and an indirect triggering of the Nrf2 pathway activation [16,40].

It is worth noting that the blood concentrations of EGCG in human volunteers were reported to range between sub-micromolar to less than 10 mM upon the intake of EGCG, EGCG-containing supplements or green tea [41,42,43,44]. The literature also reports PG conjugates in plasma reaching 2.6 mM after drinking black tea [25]. To what extent these conjugated forms will be deconjugated within tissues remains to be elucidated. It has been reported in several mice studies that feeding with diets containing 0.5 to 1% GTE or dosing at a level of 200 mg/kg/body weight (bw) green tea polyphenols can achieve in vivo Nrf2 induction in the liver, and Nrf2-dependent gene expression, e.g., Gcl, Hmox-1, can be upregulated to obtain a hepatoprotective effect [8,19]. However, not all health-beneficial effects of green tea catechins are Nrf2 dependent. For instance, Li and colleagues conducted a study on the protective effects of catechin-rich green tea extract (GTE) against high-fat-diet-induced non-alcoholic steatohepatitis in wild-type and Nrf2-knock-out mice [45]. They revealed that the anti-inflammatory and hypolipidemic activities of GTE during non-alcoholic steatohepatitis likely occur through an Nrf2-independent mechanism [45]. Moreover, the concentrations of EGCG and its microbial metabolite PG in the intestinal tract likely amount to higher levels compared to their systemic concentrations and may reach levels that are able to exert local bioactivities, including Nrf2 signaling activation in intestinal cells, which remains a topic of interest for further studies.

When EGCG is ingested, the ester bond that links the D ring is rapidly cleaved by intestinal microbiota, which liberates GA from EGCG [6]. GA was inactive at all the concentrations tested in the current study. However, it is worth noting that this compound was identified as an Nrf2 inducer in our previous work, when the reporter gene assay was performed in the absence of l-ascorbic acid [6], while all the exposure media in the present study were with freshly made 0.5 mM l-ascorbic acid. To check the difference, Nrf2 CALUX cells exposed to GA in the absence of l-ascorbic acid were tested, and a concentration-dependent increase in luciferase expression was observed (Appendix A). The decision to include l-ascorbic acid for co-exposure in this study was not only due to the fact that l-ascorbic acid is highly abundant in human plasma, but it is also able to improve the stability of polyphenols [27,28,46].

Nrf2 is a master regulator that lies at the center of the complex cellular regulatory network, mediating a vast array of cytoprotective genes. While most research schemes focus on the role of single or limited genes or proteins, such information cannot effectively reflect the molecular functional network as a whole. In this study, with the help of proteome technics, we were able to monitor the differences in the expression levels of a substantial amount of proteins between the control and different chemically exposed groups. Among EGCG and the metabolites tested, especially PG appeared able to induce significant changes to Nrf2-mediated gene expression at the protein level. Glutathione metabolism, metabolism of xenobiotics by cytochrome P450 and drug metabolism-cytochrome P450, which are three well-accepted Nrf2-related pathways that are involved in cellular antioxidant and detoxification mechanisms, were the top three most significantly altered pathways after 24 h of exposure to 30 µM PG. In the meantime, these three pathways were also among the top ten most significantly enriched pathways observed upon treatment of the cells with the reference compound t-BHQ. These results support the theory that Nrf2 pathway activation regulates detoxication and antioxidant enzymes [15,18,31,34]. For example, the rate-limiting entities in glutathione biosynthesis, glutamate-cysteine ligase catalytic subunit (Gclc) and glutamate-cysteine ligase modifier subunit (Gclm) were upregulated 1.6- and 2.3-fold after the PG treatment and 1.9- and 3.5-fold after the t-BHQ treatment. Other thiol-reducing enzymes—e.g., thioredoxin reductase 1 (Txnrd1)—were also positively regulated after PG or t-BHQ treatment.

When we further combined the proteins in the three pathways and those from the hub proteins of PG and t-BHQ treatments, a list of 22 common proteins was the result (Appendix A). Although it has been estimated that over 200 proteins are Nrf2 mediated [34], we conclude that the stimulation of the Nrf2 pathway with different chemicals may result in different protein profile fingerprints. Secondly, most of the protein level changes in PG or t-BHQ treatments were relatively mild, with FCs smaller than 1.5. There were only a few exceptions, such as Nqo1, which appeared to be the most upregulated Nrf2-mediated protein, both after PG and t-BHQ treatments, with an FC of 3.1 and 4.3, respectively. It is one of the major quinone reductases of mammalian cells and generally recognized as an extremely effective cellular detoxification enzyme that can catalyze the two-electron reduction in quinones to generate hydroquinones. In some cases, the hydroquinones are more stable and easier to be conjugated and excreted, thereby ameliorating endogenous perturbations [47]. Moreover, Nqo1 can also exert its protective functions, for example, via the generation of vitamin E and ubiquinone and inhibition of 20S proteasomal degradation [47,48]. It is considered as a cellular redox switch that is able to ameliorate oxidative stress by multiple mechanisms [47,49].

The Nrf2 activation builds an interface between the cellular redox and the intermediary metabolism. It regulates the NADPH synthesis and utilization [34,50]. This was corroborated by the results of the present study, indicating that all the four reported NADPH-generating enzymes (G6pdx, Pgd, Me1 and Idh1) were upregulated after PG and t-BHQ treatments. NADPH can serve as the cofactor for various drug-metabolizing and antioxidant enzymes, including Glutathione reductase (Gsr), Nqo1, Aldo-keto reductase (Akr) and many others [51]. NADPH is essential for them to exert their protective roles during stressed conditions. Another intermediary metabolism that has been demonstrated to be one of the major cellular regulators of redox homeostasis and biosynthesis is the pentose phosphate pathway (PPP), also known as the pentose phosphate shunt [52]. This pathway was enriched only in PG and t-BHQ treatments, once again suggesting the potent bioactivity of PG and t-BHQ in the activation of the Nrf2 pathway. The upregulated G6pdx and Pgd (due to Nrf2 activation) facilitated the oxidative branching of the PPP, thereby directing the carbon flux toward the PPP instead of glycolysis [52]. Importantly, the oxidative branch of the PPP is the primary contributor to the cytosolic NADPH generation in mammalian cells, playing a vital role in cellular detoxication and antioxidation [53]. Moreover, Nrf2 activation could also positively regulate the expression of Transketolase (Tkt) and Transaldolase (Taldo), two important enzymes in the non-oxidative branch of the PPP, thereby promoting the PPP [34,52]. However, it is important to mention that with the t-BHQ treatment, the Tkt and Taldo protein levels were increased only in the current study. Lastly, other physiological pathways, e.g., the mTOR signaling pathway and the insulin signaling pathway, were also significantly enriched after the PG treatment. Whether or how these pathways are involved in an Nrf2-mediated regulatory network is of interest to explore. Studies on illustrating the potential crosstalk between those pathways and the Nrf2 pathway may shed light on the sophisticated mechanisms behind the observed health-related physiological effects.

## 5. Conclusions

In conclusion, this study revealed the different characteristics of EGCG, EGC, GA and PG in terms of their ability to activate Nrf2-mediated gene expression. By using proteomic technics for the first time, we obtained an extensive picture of protein changes after the treatment of the cells with EGCG and its major microbial catechol-moiety-containing metabolites. It was shown that the microbial metabolite PG is more potent than its parent compound EGCG in inducing Nrf2 pathway activation. Certain intermediary metabolic pathways—e.g., glutathione metabolism, xenobiotics and drug metabolism and the PPP, which are positively regulated after Nrf2 pathway activation—may contribute to the cytoprotective functions of green tea polyphenols. Altogether, our results illustrate the pivotal role of intestinal microbial metabolites of EGCG in inducing Nrf2-associated beneficial gene expression. This finding may contribute to the observed health-promoting effects that have previously been attributed to EGCG or green tea consumption.

## Figures and Tables

**Figure 1 nutrients-14-03392-f001:**
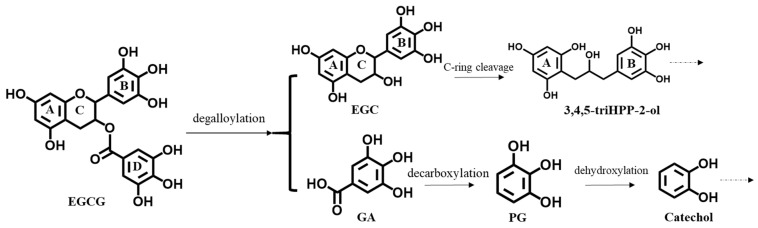
Pathways of human intestinal microbial metabolism of EGCG. EGCG, (-)-epigallocatechin gallate; EGC, (-)-epigallocatechin; GA, gallic acid; PG, pyrogallol; 3,4,5-triHPP-2-ol; 1-(3′,4′,5′-trihydroxyphenyl)-3-(2″,4″,6″-trihydroxyphenyl)-2-propanol. Dotted arrows represent further degradations.

**Figure 2 nutrients-14-03392-f002:**
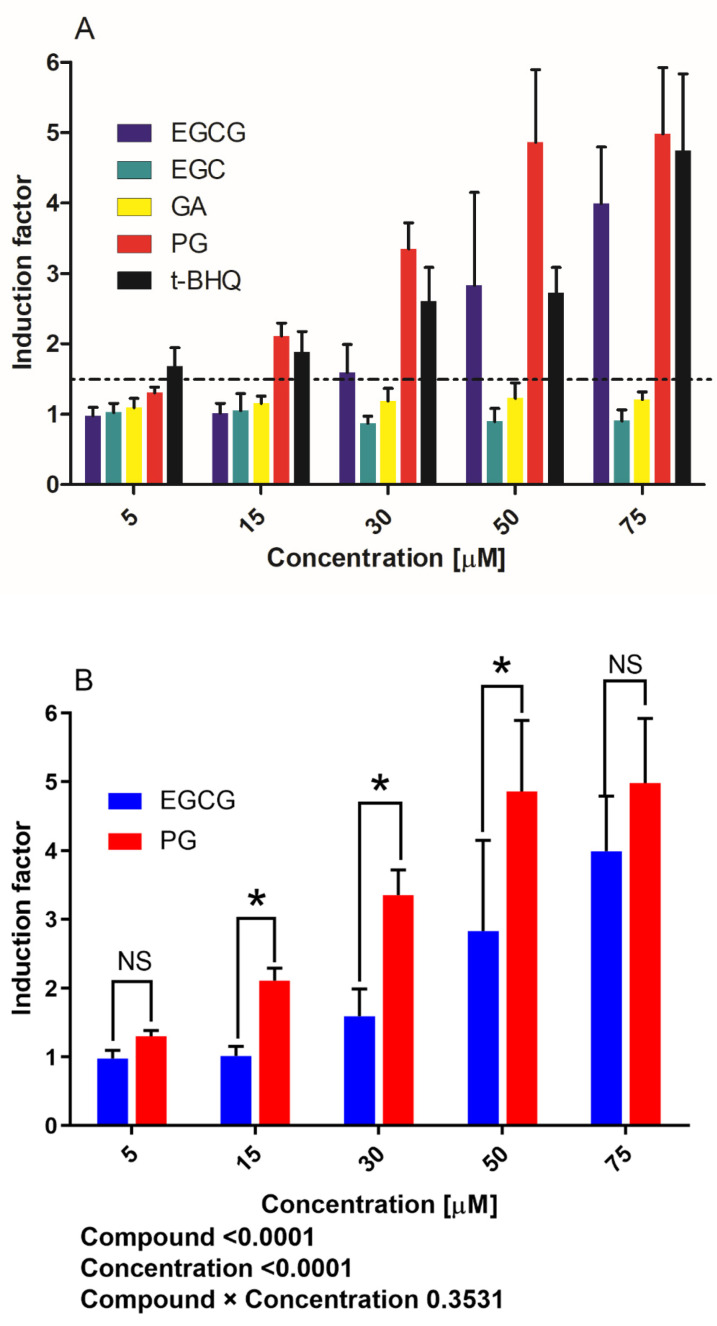
Induction of luciferase activity in U2OS-Nrf2 CALUX reporter cells after 24 h exposure to (**A**) EGCG, EGC, GA, PG or t-BHQ and (**B**) EGCG or PG at various concentrations, including a two-way ANOVA with Bonferroni multiple comparisons to evaluate the Nrf2 induction factors at the same exposure concentrations. The average IF of the positive control (20 µM curcumin) was 42.3. *p* values for main effects due to compounds and concentrations and their interaction are annotated below the figure. Statistical differences between two treatments are demonstrated (*, *p* < 0.05). (**C**) Luciferase induction in U2OS-Cytotox CALUX cells. The induction factors are presented as mean ± standard error of the mean (SEM) compared to solvent control and were derived from at least three independent experiments. EGCG, (-)-epigallocatechin gallate; EGC, (-)-epigallocatechin; GA, gallic acid; PG, pyrogallol; t-BHQ, tert-butylhydroquinone; Nrf2, nuclear factor-erythroid 2-related factor 2; IF, induction factor; NS, not significant.

**Figure 3 nutrients-14-03392-f003:**
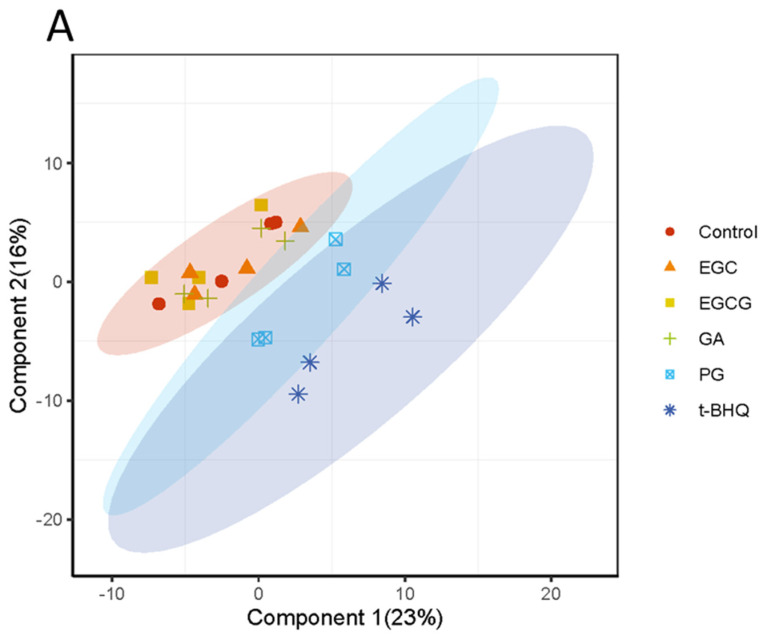
Principal component analysis (PCA) scatter plot (**A**) and heatmap (**B**) based on the expression levels or protein fold changes (FCs) of 97 Nrf2-related proteins in different treatments of Hepa1c1c7 cells. Results derived from four independent experiments. EGCG, (−)-epigallocatechin gallate; EGC, (−)-epigallocatechin; GA, gallic acid; PG, pyrogallol; t-BHQ, tert-butylhydroquinone. Full form of the abbreviated protein names can be found in Appendix A.

**Figure 4 nutrients-14-03392-f004:**
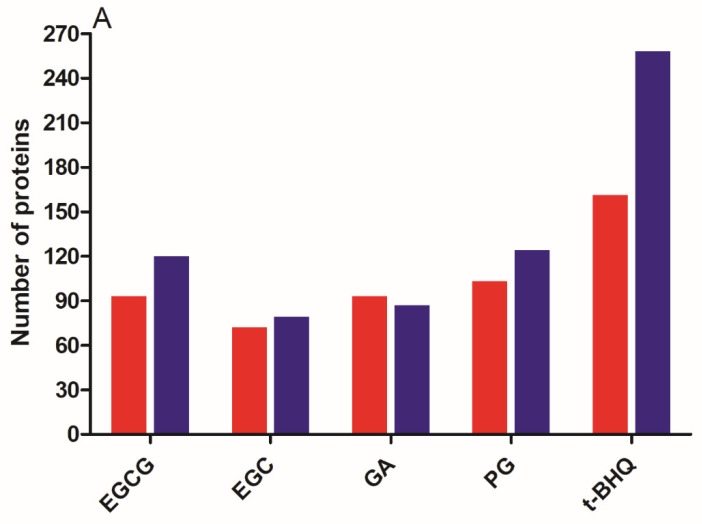
Numbers of differentially expressed proteins (DEPs) derived from different treatments (in four biological replicates) compared with the solvent control (**A**). Red and blue bars represent up- and downregulation, respectively. KEGG pathway enrichment analysis of DEPs for Hepa1c1c7 cells treated with 30 µM of PG (**B**) and t-BHQ (**C**). Only the top ten enriched terms were plotted on the graphs, and the complete results of enrichment analysis can be found in Appendix A. Results derived from four independent experiments. PG, pyrogallol; t-BHQ, tert-butylhydroquinone; KEGG, Kyoto Encyclopedia of Genes and Genomes; DEPs, differentially expressed proteins.

**Figure 5 nutrients-14-03392-f005:**
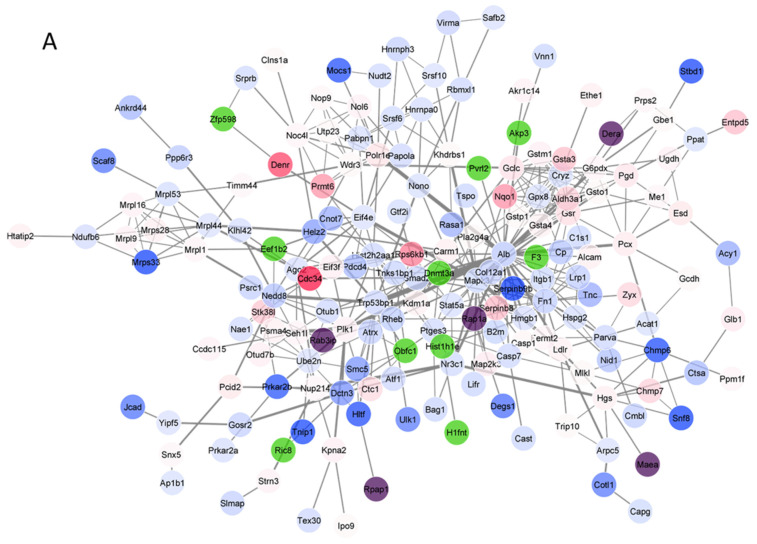
Protein–protein interaction (PPI) of differentially expressed proteins (DEPs) for the PG (**A**) and t-BHQ (**C**) treatments and hub proteins derived from the PPI of the PG (**B**) and t-BHQ (**D**) treatments. The color indicates the expression level of DEPs, with red indicating upregulation and blue indicating downregulation. Green-colored proteins were not in the DEPs but interacted with some of the DEPs; purple-colored proteins were highly induced proteins with a fold change >7. The widths of the edges were positively correlated with edge betweennesses. Results derived from four independent experiments. PG, pyrogallol; t-BHQ, tert-butylhydroquinone. Full form of the abbreviated protein names can be found in Appendix A.

**Figure 6 nutrients-14-03392-f006:**
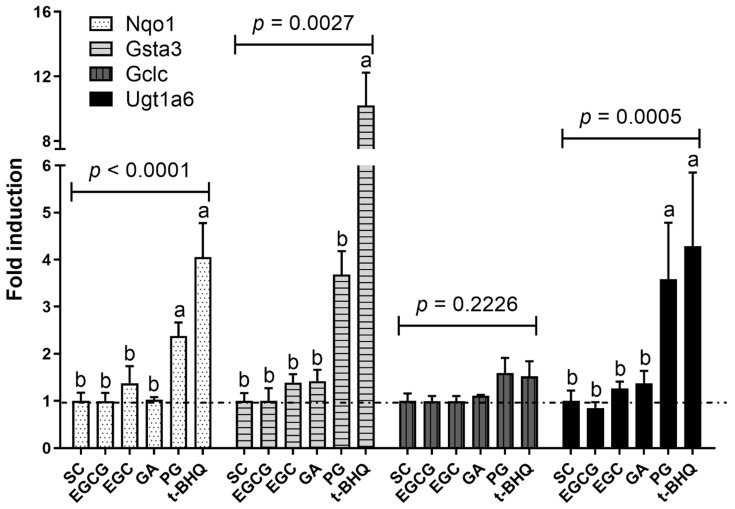
Relative RNA levels in Hepa1c1c7 cells after exposure to the selected model compounds at 30 µM. The results were calculated relative to solvent control (SC) and are presented as mean ± SEM, derived from four independent experiments. One-way ANOVA with Newman–Keuls post-test was performed. Group means not sharing a common superscript are different from each other (*p* < 0.05). Results derived from four independent experiments. EGCG, (−)-epigallocatechin gallate; EGC, (−)-epigallocatechin; GA, gallic acid; PG, pyrogallol; t-BHQ, tert-butylhydroquinone; Nqo1, NAD(P)H dehydrogenase (quinone) 1; Gsta3, Glutathione s-transferase A3; Gclc, Glutamate-cysteine ligase catalytic subunit; Ugt1a6, UDP glucuronosyltransferase family 1 member a6.

## Data Availability

All data described in the manuscript will be made available upon request pending approval by the corresponding author C.L.

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
