# Peer review of "The Gut Microbial Metabolite Pyrogallol Is a More Potent Inducer of Nrf2-Associated Gene Expression Than Its Parent Compound Green Tea (-)-Epigallocatechin Gallate"

_nutrients, 2022, doi:10.3390/nu14163392_

Round 1

Reviewer 1 Report

The manuscript has presented extensive analysis on the induction effects of various EGCG and associated metabolites in activating Nrf2 activity. Specifically, the study selected PG and t-BHQ for further analysis in terms regulating targeted proteins (using GO, KEGG and PPI analysis). 

The abstract and discussion section mainly highlights the contributing effects from PG. Suggest the authors to include description on the regulatory effects of t-BHQ in comparison with PG and if they behave similarly, and the potential implications.

Author Response

The manuscript has presented extensive analysis on the induction effects of various EGCG and associated metabolites in activating Nrf2 activity. Specifically, the study selected PG and t-BHQ for further analysis in terms regulating targeted proteins (using GO, KEGG and PPI analysis).

The abstract and discussion section mainly highlights the contributing effects from PG. Suggest the authors to include description on the regulatory effects of t-BHQ in comparison with PG and if they behave similarly, and the potential implications.

Reply: We are grateful for the reviewer’s throughout reading and positive evaluation of the manuscript. T-BHQ is well-recognized Nrf2-activator, and it has been also used in the current proteomics study as positive control. However, the aim of this manuscript is to characterize/compare the Nrf2-related gene expression induced by EGCG and its microbial metabolites. Therefore, if possible, we would like to keep the Abstract in its present form (also due to limit of 200 words in Abstract). The comparisons on regulatory effects of t-BHQ and PG, and the implications have already been included in the Discussion of this manuscript. For instance, from line 492 to 503, it compared the most significant (and commonly) enriched KEGG pathways in PG and t-BHQ treatments, indicating that glutathione metabolism is potentially crucial in the detoxication effect of both compounds. Moreover, some of the important Nrf2-mediated proteins (hub proteins) that were significantly regulated after PG and t-BHQ exposure were also compared (lines 508 - 518). The NADPH synthesis potential of PG and t-BHQ was also discussed by comparing the NADPH generating enzymes (G6pdx, Pgd, Me1 and Idh1) after respective treatments (lines 519 - 529). All in all, we agree with this reviewer on comparison of regulatory effects of PG and t-BHQ in the Discussion, while this concern has been included in the original version of the manuscript.

Reviewer 2 Report

The authors studied the effect of EGCG metabolites generated by the gut microbiota on the expression of Nrf-2 and related genes. The manuscript is well written and contains important information but there are certain below-mentioned suggestions or queries that need to be addressed:

a. There is a typographical error in the title of the manuscript; the term netabolite should be replaced by metabolite.

b. What is the possible reason for choosing specifically 6 hours anaerobic fecal incubation of EGCG as one of the criteria for testing EGCG metabolites?

c. You have added ascorbic acid while studying the effect of EGCG and other metabolites to increase their stability. What is the possible reason for not studying the effect of ascorbic acid alone on the expression of Nrf-2 and related genes? Kindly comment that whether adding ascorbic acid can modulate the action of the metabolites on ROS mediated Nrf 2 expression.

Author Response

The authors studied the effect of EGCG metabolites generated by the gut microbiota on the expression of Nrf-2 and related genes. The manuscript is well written and contains important information but there are certain below-mentioned suggestions or queries that need to be addressed:

1. There is a typographical error in the title of the manuscript; the term netabolite should be replaced by metabolite.

Reply: We thank the review for the very careful check of our manuscript. We submitted the correct spelling but it was automatically changed during process. We hoped it can be revised back in the later stage.

2. What is the possible reason for choosing specifically 6 hours anaerobic fecal incubation of EGCG as one of the criteria for testing EGCG metabolites?

Reply: Our previous study revealed EGCG was already totally converted after four hours of anaerobic incubation in the in vitro fecal model. Therefore, a total of 6-hour anaerobic fecal incubation, with multiple sampling points (i.e., t = 1, 2, 3, 4, 5, 6 h in our case), of EGCG could cover most of the metabolites formed. To avoid confusion, we have modified the “a” to “the first” in line 74 in the revised manuscript.

3. You have added ascorbic acid while studying the effect of EGCG and other metabolites to increase their stability. What is the possible reason for not studying the effect of ascorbic acid alone on the expression of Nrf-2 and related genes? Kindly comment that whether adding ascorbic acid can modulate the action of the metabolites on ROS mediated Nrf 2 expression.

Reply: We indeed have checked whether l-ascorbic acid alone is able to induce the Nrf2-mediated gene expression in the same cell line used in this manuscript (data not shown). The results show that, at least at the same concentration used in the present manuscript, l-ascorbic acid was not able to induce any Nrf2 signaling induction.